# LOTUS: EVASIVE AND RESILIENT BACKDOOR ATTACKS THROUGH SUB-PARTITIONING

## ABSTRACT

Backdoor attack poses a significant security threat to Deep Learning applications. Existing attacks are often not resilient to established backdoor detection and mitigation approaches. This susceptibility primarily stems from the fact that these attacks typically possess an unbounded or under-bounded attack scope. In other words, the trigger can cause misclassification for any input. This unbounded nature implies that the backdoored model overly emphasizes on spurious features of the trigger (e.g., only the color of a square patch), on which trigger inversion techniques can effortlessly generate effective triggers. In addition, the unbounded attack effects can be easily mitigated by backdoor removal methods. In this paper, we propose a novel backdoor attack LOTUS that is evasive and resilient by restricting the attack scope. Specifically, it leverages a secret function to separate samples in the victim class into a set of partitions and applies unique triggers to different partitions. Furthermore, LOTUS incorporates an effective trigger focusing mechanism, ensuring only the trigger corresponding to the partition can induce the backdoor behavior. Extensive experimental results show that LOTUS can achieve high attack success rate across 4 datasets and 7 model structures, and effectively evading 13 backdoor detection and mitigation techniques.

## 1 INTRODUCTION

Backdoor attack is a prominent security threat to Deep Learning applications, evidenced by the large body of existing attacks (Gu et al., 2019; Chen et al., 2017; Liu et al., 2018b; Salem et al., 2020; Turner et al., 2018) and defense techniques (Wang et al., 2019; Guo et al., 2020; Li et al., 2021a; Wu & Wang, 2021; Liu et al., 2018a). It injects malicious behaviors to a model such that the model operates normally on clean samples but misclassifies inputs that are stamped with a specific *trigger*. A typical way of injecting such malicious behaviors is through data poisoning (Gu et al., 2019; Liu et al., 2020; Bagdasaryan & Shmatikov, 2020). This approach introduces a small set of trigger-stamped images paired with the target label into the training data. Attackers may also manipulate the training procedure (Nguyen & Tran, 2020a;b; Doan et al., 2021), and tamper with the model's internal mechanisms (Liu et al., 2018b; Lv et al., 2023).

The majority of existing attacks rely on a uniform pattern (Gu et al., 2019; Chen et al., 2017; Liu et al., 2020; Turner et al., 2018) or a transformation function (Li et al., 2021c; Doan et al., 2021; Salem et al., 2020) as the trigger. These attacks tend to exhibit *unbounded* effects, meaning that the trigger (e.g., a transformation function (Doan et al., 2021; Li et al., 2021c)) is effective on any input. Such unbounded attack effects render the attacks less robust to existing backdoor detection and mitigation techniques. For instance, trigger inversion methods (Wang et al., 2019; Liu et al., 2019; Guo et al., 2020; Wang et al., 2020) aim to reverse engineer a trigger that can achieve high attack effectiveness given a set of validation samples. According to the results reported in the literature (Wang et al., 2019; Guo et al., 2020; Tao et al., 2022b), for a number of attacks, it is not hard to invert a pattern that closely resembles the ground-truth trigger and has a substantially high attack success rate (ASR). The ease of inverting injected triggers can be attributed to their unbounded attack effects since the trigger is effective on any sample. In addition, the unbounded attack effects are easily eliminated by existing backdoor mitigation methods. For example, simply fine-tuning the backdoored model using 5% of the training data can significantly reduce the ASR (Liu et al., 2018a; Li et al., 2021a; Wu & Wang, 2021). This is because the model pre-dominantly focuses on simple trigger features such as the trigger color and fails to learn its correlation with benign features.

Recent works propose sample-specific attacks (Nguyen & Tran, 2020a;b) that leverage adversarial training to encourage the model to focus on the correlation between the trigger and the input sample. These methods essentially utilize a transformation function as the trigger and applies the perturbations from other transformed images to an input sample without altering its label. While such adversarial-poisoning methods partially mitigate the unbounded attack effects, they are still *under-bounded* according to the literature (Wang et al., 2022b; Li et al., 2021a; Wu & Wang, 2021) (i.e., can be detected/removed). The reason is that adversarial-poisoning only considers individual triggers and neglects the combination of triggers (i.e., the correct trigger together with random triggers), which may still have attack effects.

In this paper, we introduce an evasive and resilient attack that constrains the attack effects through sub-partitioning. It aims to misclassify the samples of a victim class to a target class. For these victim-class samples, we divide them into sub-partitions and use a unique trigger for each partition. With such an attack design, existing defense such as trigger inversion is unlikely to find a uniform trigger as a set of samples are likely from different partitions, and hence fails to defend against our attack. In addition, we develop a novel trigger focusing technique to ensure that a partition can only be attacked by its designated trigger, not by any other trigger or trigger combinations. This is non-trivial as a straightforward data-poisoning or adversarial-poisoning alone is insufficient to bound the attack scope (i.e., causing a uniform attack effect). More details can be found in Section 4. Since the triggers are closely related to their corresponding partitions, it makes the attack more resilient to backdoor mitigation techniques.

Our contributions are summarized as follows: (1) We propose a new backdoor attack prototype LOTUS ("*Evasive and ResiLient BackdOor ATtacks throUgh Sub-partitioning*") that effectively bounds the attack scope. (2) We address a key challenge of the proposed attack, to precisely limit the scope of a trigger to its partition. As a straightforward data-poisoning or adversarial-poisoning is insufficient, we introduce a novel *trigger focusing* technique as the solution (Section 4.2). (3) We conduct an extensive evaluation of LOTUS on 4 datasets and 7 model structures. Our results show that LOTUS achieves a high ASR under a variety of settings. Our trigger focusing method effectively reduces the ASR on undesired victim classes and partitions. Furthermore, our experiments demonstrate that LOTUS is evasive and resilient against 13 state-of-the-art backdoor defense techniques, substantially outperforming existing backdoor attacks.

**Threat Model.** We follow the same threat model as state-of-the-art backdoor attacks (Doan et al., 2021; Nguyen & Tran, 2020a;b), where the adversary has control over the training procedure and provides a model to victim users after training. The adversary's goal is to achieve high attack effectiveness while also ensuring the attack's evasiveness and resilience against defense. LOTUS primarily focuses on label-specific attack. It can be easily extended to universal attack that aims to flip samples from all classes to a target class. The defender possesses white-box access to the model and a small set of clean samples for each class. She aims to determine if a model contains backdoor or mitigate the backdoor effects based on the validation samples. In our attack, the sub-partitioning function and the corresponding triggers are the secret of the attacker.

## 2 RELATED WORK

**Backdoor Attack.** As mentioned in the introduction, existing backdoor attacks use uniform patterns (Gu et al., 2019; Chen et al., 2017; Liu et al., 2020), or complex transformations (Nguyen & Tran, 2020a; Cheng et al., 2021; Nguyen & Tran, 2020b; Doan et al., 2021; Salem et al., 2020; Li et al., 2021c) to serve as the trigger. In addition, the attacker can leverage adversarial perturbations within a small bound to derive poisoned data (Shafahi et al., 2018; Zhu et al., 2019; Zhao et al., 2020; Saha et al., 2020; Wang et al., 2022c), making them indistinguishable from normal data. Subpopulation attack (Jagielski et al., 2021) is a recent data poisoning technique related to LOTUS. It is an availability attack, and its primary objective is to decrease the test accuracy of a specific subpopulation within the dataset. In contrast, LOTUS is a comprehensive backdoor attack with the intention of injecting a backdoor into the model. Therefore, these two attacks differ significantly. Subpopulation attack does not involve trigger injection or require the implementation of trigger focusing, making it distinct from LOTUS in terms of its objectives and mechanisms.

**Backdoor Defense.** Backdoor defense involves backdoor detection on model and dataset, certified robustness, as well as backdoor mitigation. Backdoor detection aims to determine whether a model

is poisoned, it can be performed by analyzing the model's behavior and output, such as Neural Clean (Wang et al., 2019), ABS (Liu et al., 2019), and so on (Wang et al., 2022b; Kolouri et al., 2020; Xu et al., 2019; Qiao et al., 2019; Guo et al., 2020; Huang et al., 2019; Tao et al., 2022b; Shen et al., 2021; Wang et al., 2020). Another type of defenses focuses on detecting poisoned data instead of models, it can be achieved through techniques such as data sanitization and outlier detection (Ma et al., 2019; Tang et al., 2021; Gao et al., 2019; Chen et al., 2018; Li et al., 2020; Liu et al., 2017; Chou et al., 2020; Tran et al., 2018; Fu et al., 2020; Chan & Ong, 2019; Du et al., 2019; Veldanda et al., 2020; Hayase et al., 2021). Certified robustness provides assurance that the the classification results are reliable and robust, even in the presence of backdoors (Xiang et al., 2021a; 2022; McCoyd et al., 2020; Jia et al., 2022). Backdoor mitigation aims to remove the backdoor effects from the attacked models (Liu et al., 2018a; Borgnia et al., 2020; Zeng et al., 2020; Tao et al., 2022a; Zhang et al., 2023; Li et al., 2021a;b; Wang et al., 2022a).

## 3    ATTACK DEFINITION

We formally define our attack in this section. For a typical classification task, given $(\boldsymbol{x}, y) \sim \mathcal{D}$ where sample $\boldsymbol{x} \in \mathbb{R}^d$ and label $y \in \{1, 2, \cdots N\}$, the goal is to train a classifier $M_\theta : \mathbb{R}^d \to \{1, 2, \cdots, N\}$, such that parameters $\theta = \arg\max_\theta P_{(\boldsymbol{x}, y) \sim \mathcal{D}}[M_\theta(\boldsymbol{x}) = y]$. Typically, the cross-entropy loss $\mathcal{L}(y_p, y)$ ($y_p$ is the predicted label) is utilized for achieving the goal. In this case, the optimization problem can be expressed as $\theta = \arg\min_\theta \mathbb{E}_{(\boldsymbol{x}, y) \sim \mathcal{D}}[\mathcal{L}(M_\theta(\boldsymbol{x}), y)]$.

Backdoor attack aims to derive a classifier $\overline{\mathcal{M}}_{\overline{\theta}} : \mathbb{R}^d \to \{1, 2, \cdots, N\}$ such that compromised parameters $\overline{\theta} = \arg\max_{\overline{\theta}} P_{(\boldsymbol{x}, y) \sim \mathcal{D}}[\overline{\mathcal{M}}_{\overline{\theta}}(\boldsymbol{x}) = y \,\&\, \overline{\mathcal{M}}_{\overline{\theta}}(\mathbb{T} \oplus \boldsymbol{x}_V) = y_T]$, in which $\mathbb{T}$ is the trigger and $\mathbb{T} \oplus \boldsymbol{x}_V$ injects the trigger to a victim input sample $\boldsymbol{x}_V$ whose label is $y_V$. Symbol $y_T$ denotes the attack target label. Backdoor attacks can be mainly classified to *universal attack* that aims to flip a sample $\boldsymbol{x}$ of any class with $\mathbb{T}$ to the target label, and *label-specific attack* that aims to flip any sample of a specific victim class to the target label. Based on trigger patterns, they can be classified to *input-independent backdoor* or *static backdoor* that has a fixed trigger pattern for all victim samples, and *dynamic trigger* that has changing patterns for different inputs. Our attack is a *label-specific dynamic backdoor attack*. Extending to other scenarios is relatively straightforward, e.g., X2X attacks (Xiang et al., 2021b; 2023), which involve multiple victim classes targeting multiple target classes using various triggers.

Assume there exists a partitioning algorithm $\mathcal{C}_n : \mathbb{R}^d \to \{p_1, p_2, \cdots, p_n\}$ that separates input samples to $n$ partitions. In our attack, victim samples (samples from the victim class) are partitioned to $n$ groups using $\mathcal{C}_n$ and each partition $p_i$ is assigned a unique trigger $\mathbb{T}_i$, such that only $\mathbb{T}_i \oplus \boldsymbol{x}_V^{p_i}$ can trigger the backdoor, where $i \in \{1, 2, \cdots n\}$ and $\boldsymbol{x}_V^{p_i}$ denotes the victim samples in the $i$-th partition. A straightforward design would follow the classic data poisoning, which aims to derive the following model parameters.

$$\overline{\theta} = \arg\min_{\overline{\theta}} \big( \underbrace{\mathbb{E}_{(\boldsymbol{x}, y) \sim \mathcal{D}}[\mathcal{L}(\overline{\mathcal{M}}_{\overline{\theta}}(\boldsymbol{x}), y)]}_{\text{Benign Utility Loss}} + \underbrace{\sum_{i=1}^{n} \mathbb{E}_{(\boldsymbol{x}_V^{p_i}, y_V) \sim \mathcal{D}}[\mathcal{L}(\overline{\mathcal{M}}_{\overline{\theta}}(\mathbb{T}_i \oplus \boldsymbol{x}_V^{p_i}), y_T)]}_{\text{Attack Target Loss}} \big) \quad (1)$$

The first loss term *Benign Utility Loss* aims to ensure the high benign accuracy of the model. The second term, *Attack Target Loss*, means that a trigger $\mathbb{T}_i$ can cause the $i$-th partition samples of the victim class $\boldsymbol{x}_V^{p_i}$ to misclassify, which is our attack goal. However, simple data poisoning cannot effectively bound the attack scope. As a result, a trigger for a particular partition can easily induce misclassifications for other partitions. That is, $\mathbb{T}_j \oplus \boldsymbol{x}_V^{p_i}$, where $i \neq j$, is miclassified to $y_T$. Besides, a trigger for a correctly-assigned partition of *non-victim* samples (samples from classes other than the victim class $\neg V$) can induce misclassification. That is $\mathbb{T}_i \oplus \boldsymbol{x}_{\neg V}^{p_i}$ is misclassified to $y_T$. Such unbounded attack effects can be attributed to the model's tendency to overfit on *naive* trigger features. For instance, when it encounters any trigger, it immediately predicts the target class without verifying if the background image aligns with the trigger according to the partitioning criteria. This overfitting issue renders the backdoored model being detected by trigger inversion techniques (Wang et al., 2019; 2022b). Moreover, these attack effects are not resilient to existing backdoor mitigation methods (Liu et al., 2018a; Li et al., 2021a).

Our objective is to establish a clear one-to-one correspondence between $\mathbb{T}_i$ and $\boldsymbol{x}_V^{p_i}$. That is, only $\mathbb{T}_i \oplus \boldsymbol{x}_V^{p_i}$ can cause misclassification. The intricate mapping criteria learned by the model make it

resilient to mitigation methods and evasive against trigger inversion as the defender is unlikely to assemble images from a specific partition. We hence aim to derive the following optimal model parameters.

$$\overline{\theta} = \arg\min_{\overline{\theta}} \left( \mathbb{E}_{(\boldsymbol{x},y)\sim\mathcal{D}}[\mathcal{L}(\overline{\mathcal{M}}_{\overline{\theta}}(\boldsymbol{x}),y)] + \sum_{i=1}^{n} (\mathbb{E}_{(\boldsymbol{x}_V^{p_i},y_V)\sim\mathcal{D}}[\mathcal{L}(\overline{\mathcal{M}}_{\overline{\theta}}(\mathbb{T}_i \oplus \boldsymbol{x}_V^{P_i}), y_T)] \right.$$

$$+ \underbrace{\mathbb{E}_{(\boldsymbol{x}_V^{p_i},y_V)\sim\mathcal{D}}\Big[ \sum_{\substack{\mathcal{T} \in \mathcal{P}(\{\mathbb{T}_1,\cdots,\mathbb{T}_n\}) \\ -\{\{\},\{\mathbb{T}_i\}\}}} \mathcal{L}\overline{\mathcal{M}}_{\overline{\theta}}(\mathcal{T} \oplus \boldsymbol{x}_{P_i},\ y_V)\Big]}_{\text{Dynamic Loss}} + \underbrace{\mathbb{E}_{(\boldsymbol{x}_{\neg V}^{p_i},y_{\neg V})\sim\mathcal{D}}[\mathcal{L}(\overline{\mathcal{M}}_{\overline{\theta}}(\mathbb{T}_i \oplus \boldsymbol{x}_{\neg V}^{P_i}), y_{\neg V})]}_{\text{Label-specific Loss}} \Big) \Big)$$

(2)

Note that compared to Equation 1, we introduce two additional terms, i.e., *Dynamic Loss* and *Label-specific Loss* in Equation 2. Intuitively, the dynamic loss controls that for a particular partition, only the corresponding trigger can cause misclassification, and any other trigger, or combination of/with other triggers shall be correctly predicted as the victim class. In particular, $\mathcal{T}$ is a subset of all possible triggers/combinations $\mathcal{P}(\{\mathbb{T}_1,\cdots,\mathbb{T}_n\})$, excluding empty $\{\}$ and $\{\mathbb{T}_i\}$. The last term, *Label-specific Loss*, ensures that only samples of the victim class can cause misclassification, even if they are from the correct partition. Here $\neg V$ denotes the classes other than the victim class. This two additional loss terms ensure LOTUS as a *dynamic and label-specific* attack, which render it evasive and resilient according to our evaluation in Section 5.3 5.4.

## 4    DETAILED ATTACK DESIGN

The overview of LOTUS is shown in Figure 1. Victim class input samples are first separated to partitions. We then apply unique triggers to samples from the corresponding partitions, whose labels are set to the target class. Data poisoning is then conducted to acquire a raw poisoned model, for which the injected triggers tend to have unbounded effects. To address this problem, LOTUS further introduces a trigger focusing step that strictly limits the attack scope of each trigger. It finally produces a trojaned model with triggers that are evasive and resilient.

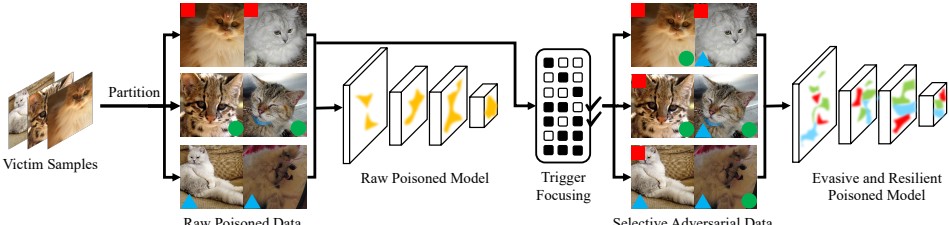

Figure 1: Overview of LOTUS

In the following, we elaborate two major components of LOTUS, namely, victim-class sample partitioning and trigger focusing.

### 4.1    VICTIM-CLASS SAMPLE PARTITIONING

LOTUS separates a set of victim-class samples into multiple partitions, and injects different triggers to different partitions. We propose two ways to partition input samples. The first is *explicit partitioning* that leverages a subset of explicit attributes of the victim class (e.g., hair color and w./ or w./o. glasses for face recognition). Assume $k$ attributes are used and each attribute has $t$ possible values. This allows to generate $t^k$ partitions. The first two columns in Figure 2 show a partitioning based on the taxonomy attribute of the bird class. Explicit partitioning leverages known attributes, which may not be available for some dataset. We hence introduce an advanced partitioning method that is applicable to arbitrary datasets in the following.

The second partitioning scheme is implicit, meaning that human uninterpretable features are used in partitioning. A straightforward idea is to directly use traditional clustering algorithms such as K-means to partition victim-class samples based on their feature representations derived from a

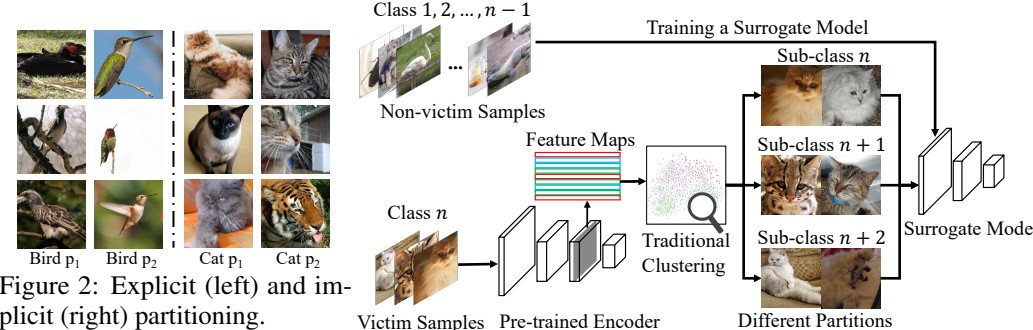

Figure 2: Explicit (left) and implicit (right) partitioning.

Figure 3: Implicit partitioning with surrogate model.

pre-trained encoder. However, according to our experiment in Appendix K.1, such a naive method does not work well. The root cause is that K-means is a clustering algorithm on a set of known data points and does not consider generalization to unseen data points. However, we need to classify a test sample to a particular cluster during attack and directly using K-means in classification does not have satisfactory results (Wu et al., 2018; Ogasawara et al., 2021; Cohn & Holm, 2021).

We hence introduce a surrogate model to help sample partitioning. Figure 3 illustrates the procedure for separating victim class samples to 3 clusters. The surrogate model has the same structure as the victim model to reduce complexity caused by structural differences. On the bottom left, the features of samples from victim class $n$ are extracted using a pre-trained encoder. We then use a traditional clustering method such as K-means to partition these samples into 3 different sub-classes based on their features. We assign labels $n$, $n + 1$, $n + 2$ to samples from the respective sub-classes. They are then combined with samples from the original classes 1 to $n - 1$ (excluding the victim class $n$) to form a new dataset consisting of $n + 2$ classes. The surrogate model is trained on this new dataset with $n + 2$ classes. The idea is to use K-means to provide a meaningful prior separation and then use classifier training to achieve generalizability. Furthermore, the decision boundaries by the surrogate model have the classes other than the victim class in consideration, whereas those by distances to centroids of K-means clusters only have victim class samples in consideration. After the training converges, the surrogate model is utilized to determine the partition of a test sample. That is, the partition index can be derived from the its classification outcome (i.e., the class with largest logits from classes $n$ to $n + 2$). The last two columns in Figure 2 show two implicit partitions of the "*cat*" class. Observe that the partitions are largely uninterpretable, which makes the attack more stealthy compared to using explicit attributes which are public.

**Handle Potential Imbalanced Examples.** We control that for any partitioning, the sizes of each partition are roughly the same, which mitigates the potential of causing partitioning bias. This is achieved by removing samples from exceptionally large clusters. In practice, such a removal is rarely needed.

## 4.2 Trigger Focusing

After partitioning, LOTUS aims to limit each trigger to its own partition, preventing it from attacking other partitions or classes. To achieve this, we design a trigger focusing technique during training.

A straightforward idea is to strictly follow the definition in Equation 2 to bound the trigger scope. However, the third term, which aims at stamping all combinations of triggers that are different from $\{\mathbb{T}_i\}$ to a sample of partition $p_i$ and setting the label to $y_V$, is extremely expensive. The number of combinations is $(2^n - 2)$, which grows exponentially with the increase of the number of partitions $n$. Moreover, the inclusion of a substantial number of additional samples will not only slow down the training but also imbalance the dataset, ultimately impacting the overall performance.

**Adversarial Poisoning Is Insufficient.** Another idea to bound the trigger scope is inspired by adversarial training (Nguyen & Tran, 2020b;a), which adds adversarial perturbations to a sample and use the original label to improve model robustness. To suppress the undesirable attack effect in our context, we could inject triggers that are not for a partition $p_i$, i.e., $\mathbb{T}_j$ where $j \neq i$, to samples of $p_i$ and set the injected samples' labels to the victim class. This approach is referred to as *adversarial poisoning*. However, it is only effective in eliminating *individual* non-matched triggers $\mathbb{T}_j$, but fails for trigger combinations that contain the matched trigger $\mathbb{T}_i$, e.g., $[\mathbb{T}_i, \mathbb{T}_j]$.

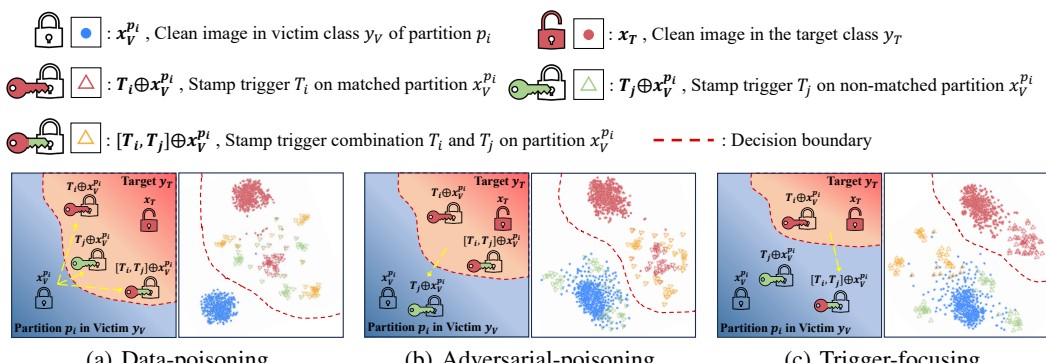

Figure 4: Decision boundaries for different poisoning strategies.

Figure 4 presents a visualization of decision boundaries for various poisoning strategies, namely: (a) Straightforward data-poisoning; (b) Adversarial-poisoning; and (c) Trigger-focusing (which will be discussed in the next paragraph). Within each subfigure, we provide an intuitive illustration and employ t-SNE (Van der Maaten & Hinton, 2008) to visualize the feature representations of different samples under these poisoning strategies. The experiment is conducted on the CIFAR-10 dataset using the ResNet18 model, and we utilize implicit partitioning to create four distinct partitions. In the figure, a hollow closed lock is used to denote clean images $x_v^{p_i}$ in the victim class of partition $p_i$, while a red opened lock is used to represent clean images of the target class. Triggers are depicted as keys with various colors. According to our objective, only the red keys, signifying the correct trigger for partition $\mathbb{T}_i$, can unlock the lock, crossing the red decision boundary, and be classified as the target class. Keys of different colors, signifying various triggers or combinations, are unable to unlock the lock and remain within the victim class region. In Figure 4(a), any trigger leads to unbounded attack effects in straightforward data-poisoning. Observe any key, denoting a trigger, can unlock the lock and cross the boundary without limitations. The t-SNE visualization on real data on the right aligns with the illustration on the left. In contrast, adversarial-poisoning, as depicted in (b), mitigates the impact of samples with unmatched individual triggers, as represented by the green key. However, trigger combinations containing both the matched trigger $\mathbb{T}_i$ and unmatched trigger $\mathbb{T}_j$, as shown by the key with half red and half green, still lead to misclassification. Similarly, in the t-SNE visualization, the yellow triangles, which represent this type of trigger combination, are substantially close to the red triangles, denoting the strictly matched triggers. This indicates the insufficiency of adversarial-poisoning.

**Efficient and Effective Trigger Focusing.** Inspired by the observation in Figure 4, we propose a novel trigger focusing method that can effectively bound trigger scopes and is in the mean time cost-effective. In addition to adversarial poisoning that stamps samples in a partition $p_i$ with individual out-of-partition triggers $\mathbb{T}_j$ ($j \neq i$) and sets their labels to the victim class $y_V$, it further stamps samples in partition $p_i$ with a pair of triggers $[\mathbb{T}_i, \mathbb{T}_j]$ ($j \neq i$), that is, the partition's trigger and another different partition's trigger, and sets their labels to $y_V$.

$$\sum_{i=1}^{n} \mathbb{E}_{(\boldsymbol{x}_V^{p_i}, y_V) \sim \mathcal{D}} \big[ \sum_{j=1, j \neq i}^{n} \big( \mathcal{L}(\overline{\mathcal{M}}_{\overline{\theta}}(\mathbb{T}_j \oplus \boldsymbol{x}_V^{p_i}), y_V) + \mathcal{L}(\overline{\mathcal{M}}_{\overline{\theta}}([\mathbb{T}_i, \mathbb{T}_j] \oplus \boldsymbol{x}_V^{p_i}), y_V) \big) \big] \qquad (3)$$

Our approach, with the dynamic loss term expressed in Equation 3, requires only $(2n - 2)$ trigger combinations, which increases linearly with the growth of partitions $n$. This number is significantly smaller than that of the dynamic loss in Equation 2.

Intuitively, the different labels of samples $\mathbb{T}_i \oplus \boldsymbol{x}_V^{p_i}$ and $[\mathbb{T}_i, \mathbb{T}_j] \oplus \boldsymbol{x}_V^{p_i}$ enable the model to learn new behaviors. As such, further stamping any other partition triggers to $[\mathbb{T}_i, \mathbb{T}_j] \oplus \boldsymbol{x}_V^{p_i}$ yields the same classification result, which is the victim class. Please refer to Appendix D for a detailed reasoning and theoretical analysis.

In Figure 4(c), it is noteworthy that trigger combinations are effectively excluded from the target class and only the trigger that matches the victim partition can cause the misclassification, well aligning with our attack objective.

## 5 EVALUATION

In this section, we evaluate on 4 benchmark datasets and 7 model structures to demonstrate the attack effectiveness of LOTUS (Section 5.2). We illustrate that LOTUS is evasive and resilient against 13 state-of-the-art detection/defense methods, compared with 7 popular backdoor attacks (Section 5.3 5.4). Besides the main results, we evaluate LOTUS against 2 poisoned sample detection baselines in Appendix G and show its evasiveness against them. We validate the effectiveness of Trigger Focusing (Section 4.2) through comparison with straightforward poisoning strategies in Appendix E. We extend LOTUS to universal attacks in Appendix H and compare LOTUS with two additional novel attacks, demonstrating its superiority over them in Appendix I. We study the effectiveness of LOTUS under adaptive defense scenarios in Appendix J. A series of ablation studies are carried out to understand the effects of different components of LOTUS in Appendix K. We also provide examples of inverted triggers in Appendix C and GradCAM visualization in Appendix L.

### 5.1 EXPERIMENT SETUP

We evaluate LOTUS on 4 widely-used benchmarks, CIFAR-10 (Krizhevsky et al., 2009), CIFAR-100 (Krizhevsky et al., 2009), CelebA (Liu et al., 2015), and restricted ImageNet (RImageNet) (Engstrom et al., 2019; Santurkar et al., 2019; Tsipras et al., 2018). Detailed description of these datasets can be found in Table 4 in Appendix A. We conduct experiments on 7 different model structures, including VGG11 (Simonyan & Zisserman, 2014), VGG16 (Simonyan & Zisserman, 2014), ResNet18 (He et al., 2016b), ResNet50 (He et al., 2016b), Pre-act ResNet-34 (PRN34) (He et al., 2016a), WideResNet (WRN) (Zagoruyko & Komodakis, 2016), and Densenet (Huang et al., 2017).

We leverage several sub-partitioning methods to partition samples from the victim class. We utilize secondary labeling, e.g., various cat species, to create clear and explicit partitions. For implicit partitioning, we first leverage K-means clustering (Hartigan & Wong, 1979) and GMM (McLachlan & Basford, 1988) to partition the feature representations of victim samples through a pre-trained encoder (Zhang et al., 2018). Then we train a surrogate model to learn the partitioning principle, which serves as the implicit sub-partitioner (Section 4.1). Details of the sub-partitioning and encoder can be found in Appendix B.

Table 1: Evaluation of attack effectiveness. The first three columns denote different partitioning algorithms (PA), datasets, and model structures. The following columns present the original accuracy of clean models (Acc.), benign accuracy of the backdoored models (BA), the attack success rate when stamping a trigger on the proper partition (ASR), and the average ASR when stamping other triggers and trigger combinations, with the standard deviation) (ASR-other).

| PA | Dataset | Model | Acc. | BA | ASR | ASR-other |
|---|---|---|---|---|---|---|
| K-means | CIFAR-10 | VGG11 | 92.16% | 92.04% | 93.80% | 4.77% ± 19.27% |
| | | ResNet18 | 95.22% | 94.71% | 94.30% | 4.39% ± 17.08% |
| | CIFAR-100 | Densenet | 75.14% | 75.15% | 92.00% | 4.36% ± 14.24% |
| | | PRN34 | 74.70% | 74.52% | 89.00% | 5.43% ± 13.50% |
| | CelebA | WRN | 80.47% | 79.40% | 92.33% | 6.87% ± 17.49% |
| | RImageNet | ResNet50 | 97.77% | 97.19% | 93.87% | 2.16% ± 19.34% |
| GMM | CIFAR-10 | ResNet18 | 95.22% | 94.59% | 90.70% | 4.80% ± 21.38% |
| | CIFAR-100 | PRN34 | 74.70% | 74.02% | 91.00% | 2.21% ± 12.57% |
| | CelebA | WRN | 80.47% | 79.66% | 92.53% | 5.39% ± 16.77% |
| | RImageNet | VGG16 | 96.51% | 95.93% | 93.52% | 3.11% ± 14.39% |
| Sec. | RImageNet | VGG16 | 96.51% | 96.36% | 96.50% | 1.79% ± 13.24% |
| | | ResNet50 | 97.77% | 97.08% | 92.50% | 2.14% ± 16.53% |

### 5.2 ATTACK EFFECTIVENESS

We evaluate the performance of LOTUS on various datasets, model structures and partitioning methods. Table 1 presents the results. For all the experiments, we use the first class of each dataset as the victim and the last class as the target. We generate 4 partitions for the victim class throughout all datasets and model structures. Our triggers are polygon patches with single colors injected on the

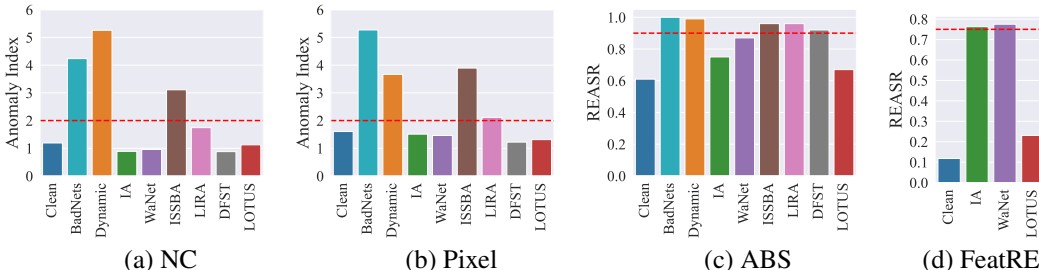

Figure 5: Evaluation of LOTUS against four trigger-inversion based backdoor detection methods.

side or in the corner of input images, which avoids occluding the features for normal classification tasks. Example images with triggers can be found in Figure 11 in Appendix. The top two blocks in Table 1 (separated by the double lines) show the results for implicit partitioning, and the bottom for explicit partitioning. For K-means clustering, ASRs are at least 89.00%, with the highest ASR of 94.30% for ResNet18 on CIFAR-10, while the degradation of benign accuracy is within 1.07%. This indicates LOTUS is a highly effective attack, which injects successful malicious behaviors to the model while maintains its benign utility. The last column shows the ASR when trigger/trigger-combinations other than a partition's trigger are stamped on the partition (ASR-other). Observe that the average ASR-other is less than 6.87%, delineating the effectiveness of trigger focusing (a trigger is only effective for the corresponding partition). A more comprehensive study on trigger focusing is presented in Section E. We have similar observations for using GMM in implicit partitioning.

For the explicit secondary labeling, LOTUS can achieve an ASR over 92.50%. The ASR-others are also quite small. The better performance of LOTUS using secondary labeling can be attributed to the fact that the victim class in RImageNet is merged from a set of similar classes in ImageNet. Those classes are naturally separable, which can be easily differentiated by the model when triggers are injected on different partitions.

Besides, we also evaluate the label specificity of LOTUS on several models. Results are presented in Table 2. Observe that even if the trigger is stamped on the proper partition of the input im-

Table 2: Evaluation of label specificity. ASR-victim means the attack success rate when stamping a trigger on the proper partition of victim class images. ASR-other-label means the attack success rate when stamping a trigger on the proper partition of other class images.

| Dataset | Network | ASR-victim | ASR-other-label |
|---------|---------|------------|-----------------|
| CIFAR10 | ResNet18 | 93.80% | 14.37% |
| CIFAR100 | Densenet | 92.00% | 11.23% |
| CelebA | WRN | 92.33% | 19.67% |
| RImageNet | VGG16 | 93.52% | 12.22% |

age, the ASR-other-label is low ($< 20\%$) because the input image is not of the victim class. The result shows that LOTUS exhibits a high level of label specificity. Furthermore, LOTUS offers an easy extension into universal attack scenarios through the integration of explicit partitioning techniques. Detailed examples can be found in Section H.

## 5.3 EVASIVENESS AGAINST BACKDOOR DETECTION METHODS

In this section, we study the evasiveness of LOTUS against 4 well-known trigger-inversion based backdoor detection methods, including Neural Cleanse (NC) (Wang et al., 2019), Pixel (Tao et al., 2022b), ABS (Liu et al., 2019), and FeatRE (Wang et al., 2022b). We compare the results of LOTUS with 7 novel backdoor attacks, including BadNets (Gu et al., 2019), Dynamic backdoor (Salem et al., 2020), Input-aware (IA) (Nguyen & Tran, 2020a), WaNet (Nguyen & Tran, 2020b), ISSBA (Li et al., 2021c), LIRA (Doan et al., 2021), and DFST (Cheng et al., 2021). For fair comparison, we launch all backdoor attacks on ResNet18 models trained on CIFAR-10. As LOTUS is a label-specific attack, we implement all other attacks in label-specific setting, where the poisoned samples are composed of images from victim class 0 stamped with the trigger and labeled as the target class 9. Besides, all detection methods are required to invert triggers based on 100 clean validation images from the victim class, targeting to labels other than it. We follow all the other settings and techniques of the original papers to implement the attack and detection methods.

Figure 5 illustrates the detection results, where the x-axis denotes different attacks and the y-axis denotes the decision scores of each baseline. The thresholds are highlighted in red dashed lines. If the decision score of an attack is higher than the threshold, it's considered to be backdoored by the

Table 3: Evaluation of resilience against backdoor mitigation methods. The first column denotes the attacks, with the following columns representing the performance of different mitigation methods. A resilient attack is expected to have both high benign accuracy (BA) and ASR after mitigation. The best results are in bold.

| Attacks | w./o. Mitigation | | Fine-tuning | | Fine-pruning | | NAD | | ANP | |
|---|---|---|---|---|---|---|---|---|---|---|
| | BA | ASR | BA | ASR | BA | ASR | BA | ASR | BA | ASR |
| BadNets | 92.02% | 100.00% | 89.31% | 1.74% | 91.70% | 0.53% | 87.81% | 0.80% | 89.15% | 0.32% |
| Dynamic | 91.81% | 100.00% | 88.87% | 2.91% | 91.39% | 22.03% | 89.11% | 2.90% | 88.25% | 12.81% |
| IA | 91.70% | 99.65% | 87.74% | 2.78% | 91.07% | 0.17% | 87.14% | 2.29% | 88.73% | 1.98% |
| WaNet | 91.22% | 98.57% | 89.56% | 1.37% | 90.22% | 1.07% | 89.74% | 1.40% | 89.07% | 0.54% |
| ISSBA | 91.67% | 99.96% | 87.73% | 2.72% | 91.12% | 14.27% | 87.97% | 2.83% | 85.64% | 10.01% |
| LIRA | 91.70% | 100.00% | 89.96% | 2.19% | 91.29% | 12.14% | 90.23% | 2.32% | 89.70% | **37.91%** |
| DFST | 91.81% | 99.97% | 88.49% | 22.86% | 91.47% | 21.61% | 88.52% | 24.66% | 87.13% | 36.17% |
| LOTUS | 91.54% | 93.80% | 88.10% | **46.90%** | 91.14% | **44.90%** | 87.61% | **42.30%** | 88.14% | 34.90% |

baseline. Specifically, NC (Wang et al., 2019) and Pixel (Tao et al., 2022b) use anomaly index as their decision scores while ABS (Liu et al., 2019) and FeatRE (Wang et al., 2022b) leverages REASR, namely the ASR of reverse-engineered triggers. Observe that NC, Pixel, ABS are effective against several attacks, including BadNets, Dynamic, ISSBA, LIRA and DFST, while leaving other advanced attacks, i.e., WaNet, IA and LOTUS. FeatRE, on the other hand, observes internal linear separability properties of existing backdoors and improves the trigger inversion process, which is able to detect the advanced backdoors operating in the feature space. Figure 5(d) shows that it can detect both IA and WaNet, but still fails to detect LOTUS. This illustrates that LOTUS is more evasive than all these baseline attacks. The underlying reason is that LOTUS leverages partitioning secrets and trigger focusing, which breaks the linear separability assumption. Without knowledge of partitioning, it's unlikely to invert a trigger with high ASR, and hence unlikely to detect the backdoor. Examples of inverted triggers can be found in Appendix C.

We also test LOTUS in the **adaptive defense** scenario, where the defender can create partitions before detection. The results in Appendix J demonstrate that LOTUS is resilient against adaptive defense strategies, as guessing the correct partitioning is challenging.

Besides trigger inversion methods, we also evaluate LOTUS using meta-classifiers, e.g., MNTD (Xu et al., 2019) and ULP (Kolouri et al., 2020), which train model-level classifiers for detection. Results in Appendix F show that LOTUS is evasive against them.

## 5.4 RESILIENCE AGAINST BACKDOOR MITIGATION METHODS

In this section, we study the resilience of LOTUS against 4 state-of-the-art backdoor mitigation methods, including standard Fine-tuning, Fine-pruning (Liu et al., 2018a), NAD (Li et al., 2021a), and ANP (Wu & Wang, 2021). We compare the results of LOTUS with 7 novel backdoor attacks. For fair comparison, all the models are trained using VGG11 on CIFAR-10 dataset. For each mitigation method, we assume the access to 5% of the training data. Besides, some standard input argumentation techniques are used, e.g., random cropping and horizontal flipping. We follow the original setting to conduct these baseline methods.

Table 3 provides the result. Observe that for all the baselines, benign accuracy change is slight, meaning that the mitigation preserves the model utility on benign tasks. However, ASR degradation is considerable for all backdoored models. Note that LOTUS can still remain part of the attack effectiveness with 34.90%-46.90%, outperforming all other attacks. The result indicates that LOTUS is more resilient against baseline mitigation methods compared to the existing attacks. This can be attributed to the design that LOTUS learns the correlation between the partitions and triggers which is hard to unlearn. Other attacks only learn partial trigger patterns that tend to be mitigate.

## 6 CONCLUSION

We propose a novel backdoor attack that leverages sub-partitioning to restrict the attack scope. A special training method is designed to limit triggers to only their corresponding partitions. Our evaluation shows that the attack is highly effective, achieving high attack success rates. Besides, it is evasive and resilient against state-of-the-art defense techniques.

ETHICS STATEMENT

In this paper, our research does not involve human subjects, data set releases, considerations related to discrimination, bias, or fairness, and we also do not encounter legal compliance or research integrity issues. Backdoor attacks are designed to cause any inputs marked with a specific pattern to be misclassified as a target label. Consequently, backdoors are emerging as a significant security concern in real-world deployments. While malicious users could potentially exploit our method to launch attacks on pretrained models, it's important to note that, similar to adversarial attacks, our research serves the purpose of raising awareness about the existence of backdoor attacks in deep learning models. It can encourage the community to develop models that are free from backdoors and are more secure.

REPRODUCIBILITY STATEMENT

The implementation code will be released upon publication. All datasets and code platform (PyTorch) we use are public. In addition, we also provide detailed experiment setups in the Appendix.

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

APPENDIX

## A   DETAILS OF USED DATASETS

Table 4 provides the detailed description of datasets used in the experiments. We use the standard datasets of CIFAR-10 (Krizhevsky et al., 2009) and CIFAR-100 (Krizhevsky et al., 2009). For the CelebA (Liu et al., 2015), we follow the existing work (Nguyen & Tran, 2020b) to generate 8 classes based on multiple facial attributes. For the restricted ImageNet (RImageNet) dataset, following the setup in existing works (Engstrom et al., 2019; Santurkar et al., 2019; Tsipras et al., 2018), we merge similar classes into one class and generate a dataset with 20 classes. The mapping between the merged classes and their original classes is shown in Table 5. During model training, we normalize input images and perform various data augmentations, including random horizontal flipping, shifting, spinning, etc.

Table 4: Description of different datasets

| Dataset | # Classes | Shape of Images | # Training Samples | # Test Samples |
|---|---|---|---|---|
| CIFAR-10 | 10 | $32 \times 32$ | 50,000 | 10,000 |
| CIFAR-100 | 100 | $32 \times 32$ | 50,000 | 10,000 |
| CelebA | 8 | $64 \times 64$ | 162,770 | 19,867 |
| RImageNet | 20 | $224 \times 224$ | 101,837 | 3,950 |

Table 5: The mapping of classes in restricted ImageNet and class ranges in original ImageNet

| Merged Classes of RImageNet | Corresponding Original ImageNet Classes |
|---|---|
| "Birds" | 10 to 13 |
| "Turtles" | 33 to 36 |
| "Lizards" | 42 to 45 |
| "Spiders" | 72 to 75 |
| "Crabs" | 118 to 121 |
| "Dogs" | 205 to 208 |
| "Cats" | 281 to 284 |
| "Bigcats" | 289 to 292 |
| "Beetles" | 302 to 305 |
| "Butterflies" | 322 to 325 |
| "Monkeys" | 371 to 374 |
| "Fish" | 393 to 396 |
| "Fungus" | 992 to 995 |
| "Snakes" | 60 to 63 |
| "Musical-instrument" | [402, 420, 486, 546] |
| "Sportsball" | [429, 430, 768, 805] |
| "Cars-trucks" | [609, 656, 717, 734] |
| "Trains" | [466, 547, 565, 820] |
| "Clothing" | [474, 617, 834, 841] |
| "Boats" | [403, 510, 554, 625] |

## B   DETAILS OF SUB-PARTITIONING

**Secondary Labeling.** Existing datasets such as ImageNet have thousands of classes. Many classes are similar to each other (e.g., from the same species). For instance, there are at least five breeds of cats in the ImageNet dataset. We hence can leverage this to merge similar classes into one class. The breeds naturally become the partitions in the new class. This has be used in existing works (Engstrom et al., 2019; Santurkar et al., 2019; Tsipras et al., 2018). We call such a partitioning method secondary labeling. We generate a RImageNet dataset with 20 new classes. The class mapping between the

new dataset and ImageNet is presented in Table 5. For example, We use "Birds" as the victim class, which has 4 breeds (corresponding to labels 10 to 13 in the original ImageNet). Then LOTUS can directly exploit the 4 breeds to generate 4 secret partitions.

**K-means Clustering.** K-means (Hartigan & Wong, 1979) clustering aims to partition $n$ observations into $k$ clusters, where each observation is a $d$-dimensional vector. The resultant partitioning ensures that each observation belonging to a cluster has the smallest distance to its cluster center or centroid. Specifically, given a set of observations $\{x_1, x_2, \cdots, x_n\}$, K-means partitions them into $k(\leq n)$ clusters $C = \{C_1, C_2, \cdots, C_k\}$ by minimizing the within-cluster sum of distances as follows.

$$\arg\min_{C} \sum_{i=1}^{k} \sum_{x_j \in C_i} \|x_j - \mu_i\|^p, \tag{4}$$

where $\mu_i$ is the mean of observations in cluster $C_i$. We use $p = 2$ in our setting, which corresponds to the Euclidean distance.

**Gaussian Mixture Model (GMM).** A Gaussian mixture model (McLachlan & Basford, 1988) is a probabilistic model that assumes there exist a finite number of Gaussian distributions which can represent the given data points. Each Gaussian distribution denotes a cluster. Other than considering the mean of data points as in K-means, GMM also incorporates the covariance during clustering (e.g., the variance of data points within the cluster).

**Pre-trained Encoders.** We utilize the pre-trained encoders available in the LPIPS GitHub repository[1], with the VGG model serving as the default encoder structure. As prior research (Zhang et al., 2018) has demonstrated, the feature maps generated by a large pre-trained encoder can offer effective perceptual representation. Therefore, we extract input sample features via the pre-trained encoder prior to implicit partitioning.

## C ILLUSTRATIONS OF INVERTED BACKDOOR TRIGGERS

In this section, we visualize some inverted triggers using Pixel (Tao et al., 2022b) in Figure 6. The first row shows the poisoned images with trigger (beginning with the original clean version for reference.) The second row presents the pixel difference between the poisoned images and their clean versions, illustrating the trigger effect. The last row visualizes the inverted triggers by Pixel. We compare the result of LOTUS at the last column with other four attacks. Observe that the inverted triggers for other attacks are visually similar to that of the ground-truth triggers. However, the inverted trigger of LOTUS is far different from the ground-truth one, which validates that LOTUS is evasive.

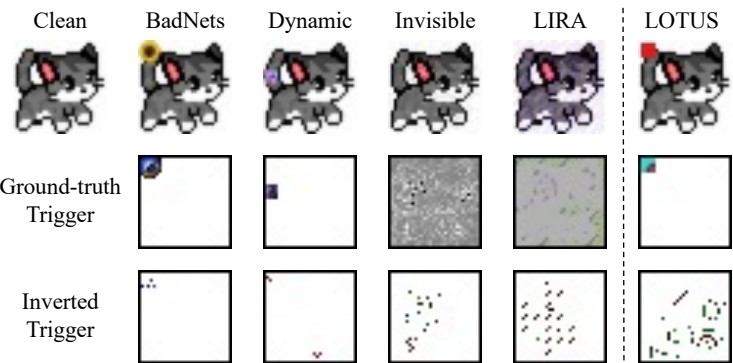

Figure 6: Visualization of inverted triggers using Pixel.

## D THEORETICAL ANALYSIS OF EFFECTIVENESS OF TRIGGER FOCUSING

Section 4.2 introduces the advantage of trigger focusing over adversarial poisoning. In this section, we formally analyze the two methods and provide our hypothesis.

---
[1]https://github.com/richzhang/PerceptualSimilarity

Poisoning a partition $p_i$ essentially introduces strong correlations between the features of $p_i$, denoted as $\mathcal{F}(p_i)$, the trigger $\mathbb{T}_i$, and the target label $y_T$, such that the conditional probability $P(y_T \mid \mathcal{F}(p_i) \cup \mathcal{F}(\mathbb{T}_i))$ is high. If another partition $p_j$ shares substantial common features with $p_i$, without any special training, $P(y_T \mid \mathcal{F}(p_j) \cup \mathcal{F}(\mathbb{T}_i))$ is high as well, meaning the model tends to predict $y_T$ when $p_j$ samples are stamped with $\mathbb{T}_i$.

**Adversarial Poisoning is Insufficient.** The goal of adversarial poisoning introduced in Section 4.2 is to unlearn the undesirable associations among features $\mathcal{F}(p_i)$, triggers $\mathbb{T}_j$ (with $j \in [1, n]$ and $j \neq i$), and the target label $y_T$. However, we observe that although $P(y_T \mid \mathcal{F}(p_j) \cup \mathcal{F}(\mathbb{T}_i))$ with $j \neq i$ becomes low after adversarial poisoning, $P(y_T \mid \mathcal{F}(p_i) \cup \mathcal{F}(\mathbb{T}_i) \cup \mathcal{F}(\mathbb{T}_j))$ and $P(y_T \mid \mathcal{F}(p_j) \cup \mathcal{F}(\mathbb{T}_i) \cup \mathcal{F}(\mathbb{T}_j))$ may still be high. It implies that stamping a set of partition triggers at the same time may still achieve a reasonably high class-wide ASR. As such, the composition of these partition triggers may constitute a uniform trigger that exceeds the attack scope. This could potentially make it vulnerable to trigger inversion techniques and various backdoor mitigation methods.

Intuitively, this is because after data poisoning, features $\mathcal{F}(p_i)$ together with the features of $\mathbb{T}_i$ become features of the target class. That is,

$$\mathcal{F}(p_i) \cup \mathcal{F}(\mathbb{T}_i) \subset \mathcal{F}(y_T)$$

In contrast, normal training tends to make $\mathcal{F}(p_i) \subset \mathcal{F}(y_V)$, that is, partition features become features of the victim class. A sample with the presence of $\mathcal{F}(p_i)$ hence already tends to be classified to $y_V$. As such, adversarial poisoning, i.e., stamping $\mathbb{T}_j$ ($j \neq i$) to samples of $p_i$, does not require the model to further learn much. The model hence tends to consider out-of-partition triggers $\mathbb{T}_j$ noises for partition $p_i$ samples, instead of considering $\mathcal{F}(p_i) \cup \mathcal{F}(\mathbb{T}_j)$ features of $y_V$. Consequently, victim class samples stamped with a trigger composition, e.g., $\boldsymbol{x}_V^{p_i} \oplus [\mathbb{T}_i, \mathbb{T}_j]$ and $\boldsymbol{x}_V^{p_j} \oplus [\mathbb{T}_i, \mathbb{T}_j]$, tend to have sufficient target class features such that they can be uniformly flipped by the combination.

We formulate the above observation with the following definition and hypothesis and then use them to explain the phenomenon.

**Definition 1** *Let $\boldsymbol{x}_V^{p_i}$ be a set of victim class samples in partition $p_i$ and $\mathcal{T}$ a subset of $\{\mathbb{T}_1, ..., \mathbb{T}_n\}$. We say $\mathcal{T}_s$ the maximum subset of $\mathcal{T}$ regarding partition $p_i$ if samples $(\boldsymbol{x}_V^{p_i} \oplus \mathcal{T}_s, y)$ have been explicitly added to the training set and have a consistent label $y$, which could be $y_V$ or $y_T$, and there is not another subset $\mathcal{T}_s'$ with $\mathcal{T}_s \subset \mathcal{T}_s' \subset \mathcal{T}$ such that $\mathcal{T}_s'$ satisfies the aforementioned condition.*

Intuitively, a *maximum subset* of a trigger set $\mathcal{T}$ regarding a partition $p_i$ is a subset that has been stamped to victim samples $\boldsymbol{x}_V^{p_i}$ and set to a consistent label. For example, in simple data poisoning, since individual triggers are only added to samples of their respective partitions. A trigger set $\{\mathbb{T}_i, \mathbb{T}_j\}$ has only one maximum subset regarding $p_i$, which is $\{\mathbb{T}_i\}$ with label $y_T$. In adversarial poisoning, the trigger set $\{\mathbb{T}_i, \mathbb{T}_j\}$ has two maximum subsets regarding $p_i$, $\{\mathbb{T}_i\}$ and $\{\mathbb{T}_j\}$. The former has the label of $y_T$ and the latter $y_V$.

**Hypothesis 1 (Maximum Trigger Subset)** *Given a trigger set $\mathcal{T} \subset \{\mathbb{T}_1, ..., \mathbb{T}_n\}$, if all the maximum subsets of $\mathcal{T}$ have a consistent label $y$, $\overline{\mathcal{M}}_{\overline{\theta}}(\mathcal{T} \oplus \boldsymbol{x}) = y$ in testing. Otherwise, the classification results are undecided.*

The hypothesis says that the *testing* results of stamping a set of triggers $\mathcal{T}$ are determined, if its maximum subsets have a consistent label *during training*. Otherwise, it is undecided. It is a hypothesis because it is difficult to quantify or formally prove. According to the hypothesis, when only simple data poisoning is used, stamping $\{\mathbb{T}_i, \mathbb{T}_j\}$ or any of its supersets to samples of partition $p_i$ in testing yields the target label $y_T$. In contrast, when adversarial poisoning is used, stamping $\{\mathbb{T}_i, \mathbb{T}_j\}$ to samples of partition $p_i$ yields an undecided label, which could be $y_T$ or $y_V$. In practice, it is more likely $y_T$. The reason is that although adversarial poisoning adds training samples $(\mathbb{T}_j \oplus \boldsymbol{x}_{p_i}, y_V)$ for each $j \in [1, n]$ with $j \neq i$. The trigger set $\{\mathbb{T}_j\}$ has a maximum subset $\{\}$ (i.e., equivalent to stamping no trigger) whose label is already $y_V$. Therefore, such additional training samples may not have substantial effects on changing model behaviors. Intuitively, the model is already capable of making the correct prediction based on $\mathcal{F}(p_i)$. It tends not to learn the additional features $\mathcal{F}(\mathbb{T}_j)$. Instead, it likely treats them as noises. Therefore, $\mathcal{F}(\mathbb{T}_i) \cup \mathcal{F}(p_i)$ dominates.

**Efficient and Effective Trigger Focusing.** Although data poisoning (i.e., setting $\mathbb{T}_i \oplus \boldsymbol{x}_{p_i}$ to $y_T$) forces features $\mathcal{F}(p_i)$ together with features $\mathcal{F}(\mathbb{T}_i)$ to become features of the target class, adding

training inputs $([\mathbb{T}_i, \mathbb{T}_j] \oplus \boldsymbol{x}_{p_i}, y_V)$ forces $\mathcal{F}(p_i) \cup \mathcal{F}(\mathbb{T}_i) \cup \mathcal{F}(\mathbb{T}_j)$ to become features of the victim class. The different labels of samples $\mathbb{T}_i \oplus \boldsymbol{x}_{p_i}$ and $[\mathbb{T}_i, \mathbb{T}_j] \oplus \boldsymbol{x}_V^{p_i}$ enable the model to learn new behaviors. As such, further stamping any other partition triggers to $[\mathbb{T}_i, \mathbb{T}_j] \oplus \boldsymbol{x}_V^{p_i}$ yields the same classification result, which is the victim class, according to the maximum trigger hypothesis.

**Hypothesis 2** *Our training method is sufficient to achieve precise focusing, meaning that only* $\mathbb{T}_i \oplus \boldsymbol{x}_V^{p_i}$ *can yield the target label and stamping any other trigger or trigger combinations yields the victim class label.*

We can prove the hypothesis assuming the maximum subset hypothesis is correct. We focus on proving that an arbitrary non-empty set of triggers $\mathcal{T} \neq \{\mathbb{T}_i\}$ must yield the victim class label for $\boldsymbol{x}_V^{p_i}$. There are two possible cases. One is when $\mathbb{T}_i \notin \mathcal{T}$ and the other is when $\mathbb{T}_i \in \mathcal{T}$. In case one, without losing generality, assume $\mathcal{T} = \{\mathbb{T}_{t_1}, ..., \mathbb{T}_{t_k}\}$ with $0 < k < (n-1)$ and $t_1, ..., t_k$ not equal to $i$. As such, $\{\mathbb{T}_{t_1}\}, ..., \{\mathbb{T}_{t_k}\}$ are the maximum subsets of $\mathcal{T}$ regarding $p_i$. They have a consistent label $y_V$. As such, $\overline{\mathcal{M}}_{\overline{\theta}}(\mathcal{T} \oplus \boldsymbol{x}_V^{p_i}) = y_V$ based on the maximum subset Hypothesis (1).

In case two, without losing generality, assume $\mathcal{T} = \{\mathbb{T}_i, \mathbb{T}_{t_1}, ..., \mathbb{T}_{t_k}\}$ with $0 < k < (n-1)$ and $t_1, ..., t_k$ not equal to $i$. As such, $\{\mathbb{T}_i, \mathbb{T}_{t_1}\}, ..., \{\mathbb{T}_i, \mathbb{T}_{t_k}\}$ are the maximum subsets of $\mathcal{T}$ regarding $p_i$. They have a consistent label $y_V$. As such, $\overline{\mathcal{M}}_{\overline{\theta}}(\mathcal{T} \oplus \boldsymbol{x}_V^{p_i}) = y_V$. The hypothesis is hence proved. $\square$

## E    EVALUATION ON POISONING STRATEGIES

Figure 7: ASR on all trigger combinations by different poisoning strategies

Table 6: Evaluation on different poisoning strategies

| Strategy | BA | ASR | ASR-indi | ASR-comb | NC index |
|---|---|---|---|---|---|
| Simple Poison | 94.79% | 98.80% | $97.86\% \pm 1.84\%$ | $97.88\% \pm 1.81\%$ | 5.338 |
| Adv. Poison | 94.47% | 94.20% | $18.95\% \pm 10.22\%$ | $73.88\% \pm 31.87\%$ | 2.161 |
| Trigger Focus | 94.71% | 91.40% | $14.15\% \pm 7.46\%$ | $0.02\% \pm 0.09\%$ | 1.156 |

We evaluate different poisoning strategies including simple data poisoning, adversarial poisoning, and trigger focusing. We employ a ResNet18 model on CIFAR-10 as the subject. The victim class is 0 and the target class is 9. We apply implicit partitioning based on K-means to generate 4 partitions. The number of possible non-empty trigger combinations is $2^4 - 1 = 15$. In the following, we use a four-bit binary to represent each combination. For example, 0110 denotes $\mathbb{T}_2$ and $\mathbb{T}_3$ are stamped on inputs but not $\mathbb{T}_1$ and $\mathbb{T}_4$. Figure 7 illustrates the ASRs on all trigger combinations by different poisoning strategies. Sub-figures from left to right present the results for simple poisoning, adversarial poisoning, and trigger focusing, respectively. In a sub-figure, each column denotes input samples

from a partition $p_i$, and each row denotes a trigger combination. The value in each cell shows the ASR when a trigger combination (row) stamped on the samples from a partition (column). Brighter the color, higher the ASR. The left sub-figure shows the ASR for simple data poisoning. Observe that all the ASRs are greater than 92.0% (with an average of 97.94%), showing the sub-partitioning is not learned by the model. The middle sub-figure is the results for adversarial poisoning. We can see around half of cells have small values, especially for single trigger combinations (the top four rows). For more complex trigger combinations, the ASRs are still high with the highest of 100.0% (trigger combination 0111 on partition $p_2$), indicating the insufficiency of adversarial poisoning. The right sub-figure is for our trigger focusing. Observe that except for stamping a trigger on the proper partition, the other cases all have a low ASR with an average of 3.04%. We compute the average ASR for individual wrong triggers (other than the correctly-assigned trigger for the partition) and trigger combinations for each strategy and report the results in Table 6. The first column denotes the poisoning strategies. The following columns show the BA (benign accuracy), ASR-indi (ASR for individual wrong triggers), ASR-comb (ASR for trigger combinations), and NC anomaly indexes, respectively. Observe that all the ASRs are almost 100% for simple poisoning. Adversarial poisoning reduces the ASR-indi to a low level while leaving ASR-comb high (73.88% on average). Our trigger focusing strategy has the lowest ASR-indi with an average of 14.15% and ASR-comb 0.02%. We further use NC (Wang et al., 2019) to evaluate on poisoned models by different strategies. The last column shows the anomaly index for different poisoned models. Observe that models poisoned by simple data poisoning and adversarial poisoning can be easily detected by NC (with anomaly index $> 2$). Poisoned models by trigger focusing, on the other hand, are able to evade NC's detection, delineating the effectiveness of trigger focusing strategy to achieve evasiveness.

## F  EVALUATION AGAINST META-CLASSIFIERS

Table 7: Evaluation on ULP and MNTD.

| # Partitions | 2 | 3 | 4 | 5 | 6 | 7 | Accuracy |
|---|---|---|---|---|---|---|---|
| ULP | 0 | 1 | 0 | 0 | 0 | 0 | 16.7% |
| MNTD | 1 | 0 | 1 | 1 | 0 | 0 | 50.0% |

In this section, we evaluate LOTUS against backdoor detection methods based on meta-classifier, i.e., MNTD (Xu et al., 2019) and ULP (Kolouri et al., 2020). Both methods aim to train a binary classifier from a large number of benign and poisoned models for backdoor scanning. They generate a set of input patterns and feeds them to the models whose output logits are then used to train the classifier. During training, the patterns and the binary classifier are optimized together in order to distinguish benign and poisoned models in the training set. During inference, these patterns are fed to the given model whose output logits are used by the binary classifier to decide whether the model is poisoned.

As MNTD and ULP require a large number of models for training, we adopt the TDC dataset[2], which consists of 125 benign models and 125 poisoned models trained on CIFAR-10 using WRN. We are able to train a MNTD and ULP classifier with over 90% training accuracy. To evaluate LOTUS against both meta-classifiers, we generate 6 poisoned models using the same dataset and structure, with 2-7 partitions. Table 7 presents the detection results by MNTD and ULP, where 0 denotes benign and 1 poisoned. Observe that the detection accuracy of ULP is only 16.7%, and MNTD 50.0%, showing they are unable to detect LOTUS.

## G  EVALUATION AGAINST TESTING-TIME SAMPLE DETECTION

We conduct experiments to study the evasiveness of LOTUS against 2 testing-time sample detection techniques, STRIP (Gao et al., 2019) and Spectral Signitures (Tran et al., 2018) on a ResNet18 model trained on CIFAR-10 using K-means implicit partitioning and a VGG16 model trained on RImageNet using secondary labeling. Specifically, for each model, we randomly select 400 clean and 400 poisoned samples and assess their distributions according to different baselines.

STRIP (Gao et al., 2019) examines the entropy of the resulting predictions to identify the presence of a backdoor trigger by superimposing an input with a set of clean samples. The overlapping distributions

---

[2]https://trojandetection.ai/

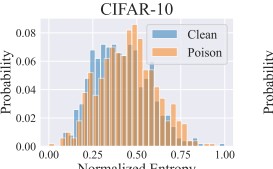 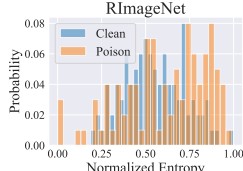 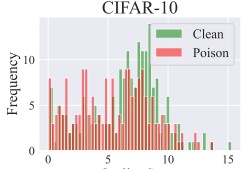 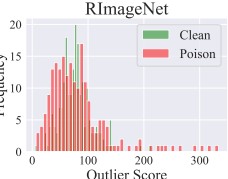

Figure 8: Evaluation against STRIP

Figure 9: Evaluation against Spectral Signitures

of the normalized entropy for clean and poisoned data in Figure 8 indicate that LOTUS evades the detection of STRIP. The reason is that super-imposing breaks condition of correct partitions to trigger the backdoor. On the other hand, Spectral Signatures (Tran et al., 2018) identifies backdoor attacks by detecting outliers in feature covariance spectra through singular value decomposition. Figure 9 visualizes the distributions of outlier scores for clean and poisoned samples, where they highly overlap, meaning Spectral Signatures is unable to detect LOTUS's poisoned samples. This is because LOTUS learns a complex mapping from partition secrets to trigger injection, making the internal values similar to that of complex benign features.

## H  EXTENSION TO UNIVERSAL ATTACKS

We extend LOTUS to universal attack by partitioning samples based on their classes. Using ResNet18 as an example model, we divide CIFAR-10 samples into 5 partitions (2 classes for each partition) and apply Trigger Focusing for target label 0. We achieve 94.14% BA, 95.99% ASR and 5.94% ASR-other. Furthermore, the attacked model successfully evades the detection by NC, with an anomaly index of 1.614, which is below the established outlier threshold of 2.

## I  COMPARISON WITH ADDITIONAL BACKDOOR ATTACKS

We reproduce another two novel attack clean-label (Turner et al., 2018) and adaptive blend (Qi et al., 2022) on CIFAR-10 and ResNet-18 with over 94% BA and 99% ASR. NC can detect both of them and Fine-pruning can reduce their ASRs to <10%. However, NC and Fine-pruning cannot detect/mitigate LOTUS according to the results in Section 5.3 and Section 5.4, which indicates LOTUS outperforms these attacks regarding both evasiveness and resilience.

## J  ADAPTIVE DEFENSE

Table 8: Evaluation on adaptive defense.

| Partition Method | Par. 0 | Par. 1 | Par. 2 | Par. 3 | MO |
|---|---|---|---|---|---|
| GT | 2.587 | 1.985 | 2.366 | 2.841 | 100% |
| Inputs | 1.178 | 0858 | 0.691 | 0.832 | 47% |
| Internels | 1.358 | 0.699 | 1.387 | 0.905 | 67% |

Table 9: Evaluation on other partitions.

| Metrics | Half 0 + half 1 | Half 2 + half 3 | Equal 4 | Random |
|---|---|---|---|---|
| NC index | 1.316 | 0.823 | 1.041 | 1.019 |
| MO | 50% | 50% | 25% | 27% |

In this section, we assess the performance of LOTUS under various adaptive defense scenarios. We operate under the assumption that defenders have knowledge about the victim class and the number of partitions implemented by LOTUS. With this information, they can independently create partitions and utilize existing defense strategies to reverse-engineer a trigger for each partition and conduct detection. To illustrate, we conduct experiments using the ResNet18 model on CIFAR-10 attacked by LOTUS, employing four partitions generated via the implicit partitioning using K-means. We leverage NC (Wang et al., 2019) as a typical baseline for detection against LOTUS on partitioned

samples. A model with anomaly index exceeding 2, as produced by NC, is considered backdoored. Table 8 displays the outcomes for different partitioning techniques. The first column denotes the partitioning methods, with "GT" implying that the defender possesses precise knowledge of the partitions, "Inputs" suggesting that the defender employs K-means to generate partitions from input samples, and "Internals" meaning the defender utilizes K-means to partition based on internal feature representations. Subsequent columns represent NC anomaly indexes generated on various partitions. The last column, "MO" quantifies the maximum overlap between generated partitions and one of the ground-truth partitions. We observe that if defenders possess knowledge of the ground-truth partitions, they have a higher likelihood of detecting LOTUS by using samples from a specific partition. However, it is often impractical for defenders to have such knowledge. Scenarios where defenders generate partitions from input or feature data are more realistic. However, even when defenders employ the same partitioning algorithm on input samples and feature representations, they face difficulties in detecting LOTUS. This is because LOTUS leverages a surrogate model to learn partitioning principles, rather than directly using K-means results. Consequently, their partitioning outcomes differ, with an MO of only 67%. Table 9 presents additional results, testing detection scenarios where samples consist of an equal mix from two partitions (Half 0 + half 1), an equal mix from all four partitions (Equal 4), or random selections from partitions. In all instances, defenders find it challenging to detect LOTUS. In conclusion, even when defenders possess prior knowledge of LOTUS, detecting the backdoor remains challenging due to the complexity of its sub-partitioning approach.

## K    ABLATION STUDY

In this section, we conduct a series of ablation studies of LOTUS on different settings and hyper-parameters.

Table 10: Results w/ and w/o surrogate models.

| Method | BA | ASR | ASR-other |
|---|---|---|---|
| K-means | 94.36% | 84.40% | 19.01% $\pm$ 39.24% |
| K-means + surrogate | 94.71% | 94.30% | 4.39% $\pm$ 17.08% |
| GMM | 94.78% | 86.38% | 20.51% $\pm$ 40.38% |
| GMM + surrogate | 94.59% | 90.70% | 4.80% $\pm$ 21.38% |

### K.1    NECESSITY OF USING SURROGATE MODELS

We verify the necessity of training a surrogate model for implicit sub-partitioning to attain high attack effectiveness with LOTUS. To validate this, we conducted experiments utilizing the ResNet18 model on CIFAR-10 and compared the attack performance achieved through traditional partitioning methods, specifically K-means and GMM, with the performance achieved when training a surrogate model for partitioning. The outcomes are presented in Table 10.

Notably, when we omit the surrogate model, the ASRs experience a noteworthy reduction, ranging from 4% to 10%. Additionally, ASRs-other exhibit an increase. This decline in performance is due to the difficulty of the attacked model to effectively learn the traditional partitioning schemes. This also highlights the necessity of training a surrogate model for sub-partitioning to achieve good attack performance.

### K.2    EFFECT OF DIFFERENT NUMBER OF PARTITIONS

We evaluate the attack's effectiveness by generating different numbers of partitions. The experiment is conducted on ResNet18 trained on CIFAR-10 using implicit sub-partitioning. The results are shown in Figure 10, with the x-axis representing the number of partitions and the y-axis indicating the percentage values for different metrics. These metrics include BA (Backdoor Accuracy), ASR (Attack Success Rate), ASR-other (Attack Success Rate of images assigned incorrect triggers), and Acc.-other (Accuracy for images with incorrect triggers).

It is noteworthy that BAs and ASRs consistently remain at high levels, showcasing the attack's effectiveness across various partition numbers, even when the number of partitions is as high as 12

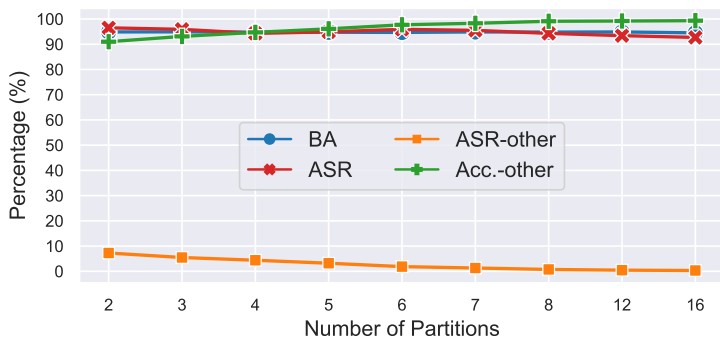

Figure 10: Attack effectiveness of different number of partitions

or 16. Additionally, ASR-other values are low, while Acc.-others values are high, indicating that images marked with incorrect triggers tend to be predicted as belonging to their source labels. This observation further indicates that LOTUS achieves good performances across various partition counts.

Table 11: Results on different number of samples per partition.

| # Samples per Partition | BA | ASR | ASR-other |
|---|---|---|---|
| 1000 (default) | 94.74% | 93.90% | 5.65% |
| 500 | 93.94% | 90.70% | 7.09% |
| 250 | 94.45% | 91.20% | 5.77% |
| 100 | 92.00% | 81.10% | 9.46% |
| 50 | 89.46% | 79.40% | 8.64% |

### K.3 EFFECT OF DIFFERENT NUMBER OF SAMPLES PER PARTITION

We explore the influence of different sample counts per partition, a crucial factor when employing sub-partitioning within the victim class. First, unevenly distributed samples resulting from partitioning could introduce fairness issues in learning. To mitigate this, we analyze the post-partitioning sample counts and strive to maintain a roughly balanced distribution, although sample imbalance rarely occurs (as discussed in Section 4.1).

Another potential challenge arises when the number of samples per partition is limited, potentially affecting trigger focusing due to the requisite knowledge of sub-partitioning secrets. To examine this effect, we conducted experiments using ResNet18 models on the CIFAR-10 dataset, generating 4 partitions within victim class 0. The results are displayed in Table 11, with the first column representing the number of samples per partition, followed by benign accuracy (BA), attack success rate (ASR), and attack success rate for non-targeted triggers (ASR-other). We compare LOTUS's performance with the default setting of 1000 samples per partition and other configurations. It is notable that as the number of samples per partition decreases, both BA and ASR exhibit declines, while ASR-other experiences an increase. These results indicate that LOTUS's performance degrades with fewer samples per partition. However, it's worth highlighting that even with just 250 samples, an ASR of over 91% can be achieved. This suggests that LOTUS generally requires around 1000 samples per victim class for good performance, which is often practical across various datasets.

Table 12: Results on different victim-target class pairs.

| Victim | Target | BA | ASR | ASR-other |
|---|---|---|---|---|
| 0 | 9 | 95.01% | 91.40% | 4.65% $\pm$ 16.06% |
| 4 | 5 | 94.94% | 93.10% | 3.64% $\pm$ 18.74% |
| 1 | 6 | 95.13% | 95.70% | 5.44% $\pm$ 22.67% |
| 7 | 4 | 95.03% | 93.40% | 5.89% $\pm$ 23.55% |
| 3 | 2 | 95.21% | 89.60% | 6.38% $\pm$ 24.44% |

### K.4 EFFECT OF DIFFERENT VICTIM-TARGET CLASS PAIRS

In the previous experiments, we use the first class as the victim and the last class as the target as the default setting. Here, we randomly select a few victim-target class pairs to study how they affect the performance of LOTUS. We conduct an experiment on ResNet18 on CIFAR-10 and use implicit sub-partitioning to generate 4 partitions. Table 12 shows the results. Observe that LOTUS consistently has a high ASR with different victim-target class pairs, and the ASR-other values are relatively low, delineating the generalizability of LOTUS to different class pairs.

Table 13: Results on different trigger patterns.

| Trigger Pattern | BA | ASR | ASR-other |
|---|---|---|---|
| Color Patch | 95.01% | 91.40% | $4.65\% \pm 16.06\%$ |
| Logo | 94.89% | 91.40% | $4.63\% \pm 16.00\%$ |
| Instagram Filter | 94.40% | 89.10% | $6.90\% \pm 21.24\%$ |

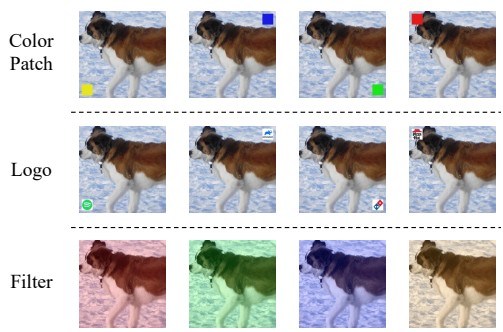

Figure 11: Different trigger patterns. The first row shows the color patch triggers, the second row logo triggers and the third row Instagram filter triggers.

### K.5 EFFECT OF DIFFERENT TRIGGERS

We use solid polygon patches as triggers in our previous experiments. We study two other types of triggers, logos and Instagram filters. Figure 11 shows different trigger patterns. The first row shows the color patch triggers, the second row the logo triggers (downloaded form the Internet), and the third row the Instagram filter triggers. We use a ResNet18 model on CIFAR-10 with 4 partitions for the study. Table 13 presents the results. All three studied cases have high BAs, ASRs, and low ASRs-other. Due to the overlapping of triggers during trigger focusing, there is a slight degradation of the attack effectiveness on Instagram filter, with 2% ASR degradation and 2% ASR-other increase. The Instagram filter cases perturb the entire image and hence slightly degrade the performance of trigger focusing. Overall, LOTUS is effective with different types of triggers.

Table 14: Results on different patch sizes.

| Patch Size | BA | ASR | ASR-other |
|---|---|---|---|
| $3 \times 3$ | 94.80% | 93.90% | 4.32% |
| $6 \times 6$ | 94.89% | 94.30% | 4.39% |
| $10 \times 10$ | 94.40% | 94.40% | 5.09% |

### K.6 EFFECT OF DIFFERENT PATCH TRIGGER SIZES

We study the effect of different trigger sizes using solid patches as the trigger. We conduct experiments on ResNet18 model on CIFAR-10 with 4 secret partitions. We evaluate patch sizes of $3 \times 3$, $6 \times 6$ and $10 \times 10$. Results are shown in Table 14, where the first column denotes the trigger sizes with the subsequent columns illustrating LOTUS's performance, i.e., BA, ASR and ASR-other. Observe that LOTUS is generally effective using multiple trigger sizes, with high benign accuracy, ASR and low ASR-other.

Table 15: Results on different pre-trained encoders.

| Encoder | BA | ASR | ASR-other |
|---|---|---|---|
| VGG | 94.71% | 94.30% | 4.39% ± 17.08% |
| AlexNet | 95.10% | 92.70% | 4.59% ± 20.92% |
| SqueezeNet | 94.75% | 91.60% | 4.26% ± 20.21% |

### K.7 EFFECT OF DIFFERENT PRE-TRAINED ENCODERS

In the previous experiments, we leverage the pre-trained encoders of VGG (Zhang et al., 2018) to extract features of victim samples. We study other two encoders of different structures. We conduct experiments on a ResNet18 model on CIFAR-10 with 4 implicit partitions. Table 15 shows the results of using different pre-trained encoders. Observe that LOTUS achieves a consistent good performance through out all encoders.

## L EVALUATION AGAINST GRADCAM

The GradCAM (Selvaraju et al., 2017) method is commonly used to visualize the important regions of inputs through gradient propagation. In this study, we evaluate LOTUS using GradCAM and compare the results with those of BadNets (Gu et al., 2019) and Dynamic (Salem et al., 2020) backdoors. Figure 12 displays the results, which are organized into four groups of images representing the GradCAM visualizations for the clean model, the BadNets attacked model, the Dynamic attacked model, and the LOTUS attacked model. In each group of images, the first row shows the poisoned images (clean images for the clean model), while the second row shows the GradCAM visualizations. The reddish regions represent important areas, while the bluish regions represent less important parts.

Our observations revealed that for both BadNets and Dynamic backdoors, the important regions are located at the trigger positions, indicating that their triggers have notable features regarding the gradients. However, for LOTUS's poisoned images, the important regions are more similar to those of the clean models. This further validates the evasiveness of LOTUS and explains why it is difficult to be inverted.

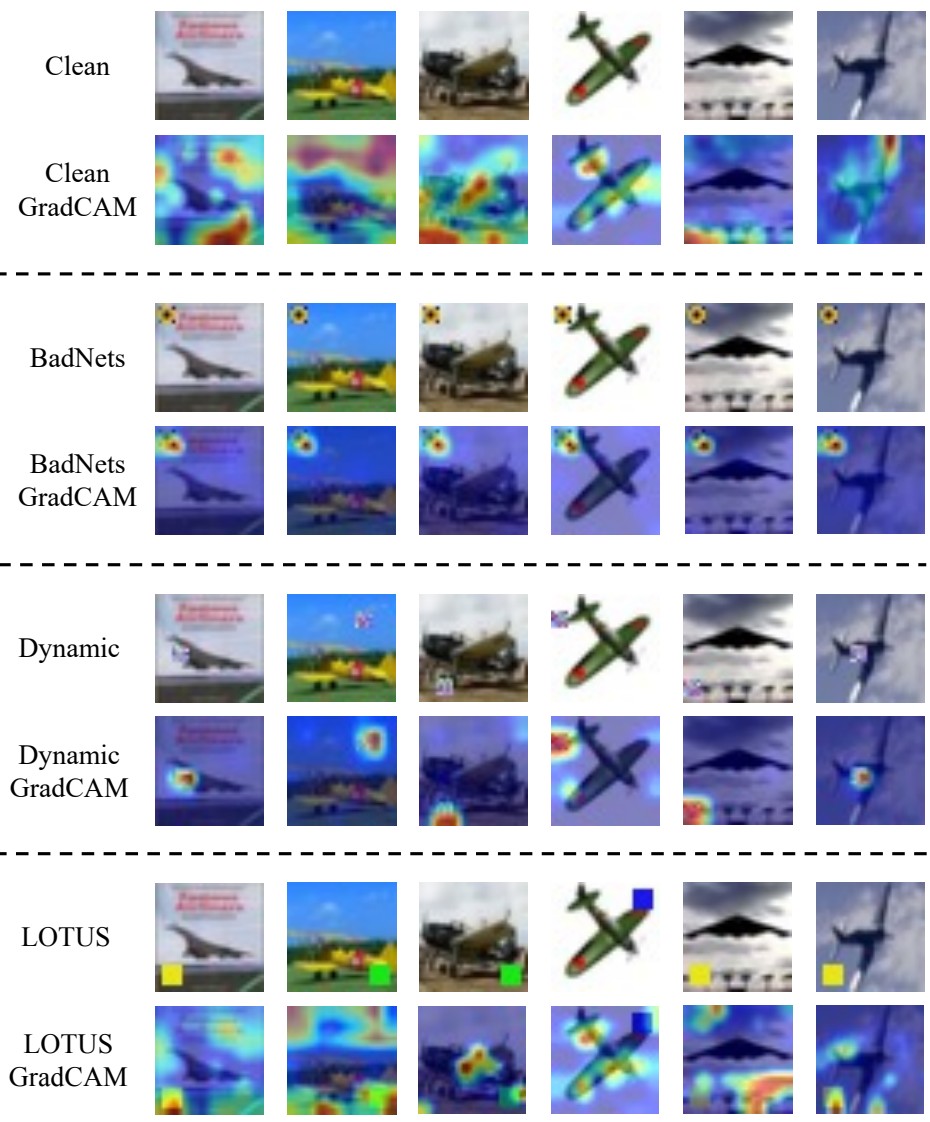

Figure 12: Important region visualizations via GradCAM.

