# OpenReview forum: "LOTUS: Evasive and Resilient Backdoor Attacks through Sub-Partitioning"
_ICLR.cc/2024/Conference — ICLR 2024 Conference Withdrawn Submission_

### Official Review · Reviewer_Mwh5 · 2023-10-30

**Soundness:** 2 fair
**Presentation:** 1 poor
**Contribution:** 2 fair
**Rating:** 3
**Confidence:** 4

**Summary:**

This paper explores how to design an adaptive attack to circumvent backdoor defenses. Specifically, the authors propose to partition selected samples into different parts where each part has a unique trigger pattern. In particular, the authors design a trigger-focusing module to ensure that a partition can only be attacked by its designated trigger, not by any other trigger or trigger combinations. The authors evaluate their method under 13 backdoor defenses on 4 datasets.

**Strengths:**

1. The topic is of great significance. Figuring out how to design more stealthy backdoor attacks (against backdoor defenses) is one of those constant themes in the field.
2. The main idea is easy to follow.
3. The authors try to analyze the resistance to adaptive attacks and why their method is effective. It should be encouraged.

**Weaknesses:**

1. In general, the writing is poor.
- The concept of bounded and under-bounded is unnecessary and misleading. Specifically, the 'bounded attack' means that the trigger is effective on any input, whereas the 'under bounded' means that attacks can be removed. The definitions of them are unrelated or even inclusive. I don't understand what purpose the author had in making those two words.
- At the beginning of page 2, the authors claim that sample-specific attacks leverage adversarial training to encourage the model to focus on the correlation between the trigger and the input sample. However, to the best of my knowledge, none of them using adversarial training. They only claim that different poisoned samples contain different trigger patterns.
- The definition part of Chapter 3 is also very confusing and redundant.
- The design part of the method is also strange. It appears that the authors only select samples from a particular category and poison them, yet the authors don't state why. In principle, shouldn't the author select samples from all categories to poison? After all, as long as multiple triggers can activate the backdoor separately and independently, the backdoor is difficult to recover?
- In addition, the authors have always emphasized that reducing the generalizability of triggers makes them more difficult to recover. I think that's right, but why not just reduce generalizability and instead still implant multiple different triggers?
- Overall, the design of the authors' methodology is not consistent with their motivation (bypass existing trigger inversion). In my opinion, it's not as straightforward as just planting a number of different backdoors at the same time.
- The caption of Figure 4 is too long.

2. Missing important technical details.
- Please provide more details about why you need to do the clustering.
- Please provide the poisoning rates of all attacks. Are they the same?

3. Please provide more details and explanations about why your attack is more effective under backdoor-removal attacks. To me, this doesn't seem to have a direct causal relationship to your method itself.

4. Missing important experiments.
- There is no ablation study about the trigger focusing module.
- The authors claim that their designed modules are effective in preventing backdoor detection. However, the authors do not verify it in their ablation study.
- Please exploit more advanced method instead of NC to design adaptive defenses. As we all know that NC is ineffective under many
cases.

**Questions:**

Please refer to the weakness part.

---

### Official Review · Reviewer_rKsD · 2023-10-31

**Soundness:** 2 fair
**Presentation:** 3 good
**Contribution:** 2 fair
**Rating:** 6
**Confidence:** 3

**Summary:**

The paper addresses the security threat posed by backdoor attacks in Deep Learning applications. Existing backdoor attacks are often susceptible to detection and mitigation techniques due to their unbounded or under-bounded attack scope, where triggers can cause misclassification for any input. This paper proposes a novel backdoor attack called LOTUS, which is evasive and resilient by limiting the attack scope. LOTUS uses a secret function to separate samples in the victim class into partitions and applies unique triggers to each partition. It also incorporates a trigger-focusing mechanism to ensure that only the trigger corresponding to a specific partition can induce the backdoor behavior. Extensive experiments demonstrate that LOTUS achieves high attack success rates across four datasets and seven model structures while effectively evading 13 backdoor detection and mitigation techniques.

**Strengths:**

This work could be attributed to the class of “controllable adversarial attack”, which explores the potential to bound the backdoor attack and enhance its stealness, which could lead to more robust models.
The strengths of this paper are as follows:
1. Originality: The paper presents a novel backdoor attack called LOTUS ("Evasive and ResiLient BackdOor ATtacks throUgh Sub-partitioning"), which effectively bounds the attack scope by dividing victim-class samples into sub-partitions and using unique triggers for each partition. This innovative approach sets it apart from existing backdoor attacks that rely on uniform patterns or complex transformations as triggers.
2. Quality: The authors address a key challenge in implementing LOTUS by introducing a novel trigger focusing technique to ensure that a partition can only be attacked by its designated trigger, not by any other trigger or trigger combinations. The paper's methodology is well-thought-out and thoroughly explained, demonstrating the quality of the research.
3. Clarity: The paper is well-written and well-organized, providing clear explanations of the concepts and techniques involved in the proposed LOTUS attack. The authors also provide a comprehensive overview of the related work, which helps contextualize their contribution within the broader field of backdoor attacks and defense.
4. Significance: The extensive evaluation of LOTUS on four datasets and seven model structures demonstrates its effectiveness in achieving high attack success rates while evading 13 state-of-the-art backdoor defense techniques. This result highlights the significance of the proposed attack and its potential impact on the security of deep learning applications.

**Weaknesses:**

While this paper introduces the novel LOTUS backdoor attack and demonstrates its effectiveness across various datasets and model structures, there are some weaknesses that could be addressed to improve the work:
1.  Extension to universal attacks: The paper mentions that LOTUS can be extended to universal attacks, but does not provide experimental evidence or a detailed discussion on how this can be achieved. It would be useful to provide more information on how LOTUS can be adapted for universal attacks and to demonstrate its effectiveness in this setting.
2. Potential countermeasures: The paper could discuss possible countermeasures that could be developed to defend against LOTUS, as well as the challenges in creating such countermeasures. This would provide a more balanced view of the attack and contribute to the ongoing research in backdoor defense.

**Questions:**

1. How does LOTUS perform in scenarios where the defender has partial knowledge of the sub-partitioning function or the corresponding triggers? Would this compromise the attack's evasiveness and resilience?
2. The paper focuses on label-specific attacks; can you elaborate on how LOTUS can be extended to universal attacks? What challenges might arise in this extension, and how do you plan to address them?

---

### Official Review · Reviewer_1k7k · 2023-11-01

**Soundness:** 3 good
**Presentation:** 3 good
**Contribution:** 3 good
**Rating:** 6
**Confidence:** 3

**Summary:**

This paper proposes a two-step approach for trojan injection into machine learning models. First, it suggests partitioning the training data and then injecting distinct triggers for each partition with respect to a specific victim class. Additionally, a novel loss function is introduced to ensure that only the trigger associated with the partition can induce the trojan behavior. The primary motivation behind this approach is to reduce the visibility of the trigger pattern, making it harder to detect trojans using existing methods.

**Strengths:**

Strengths:

- The evaluation demonstrates that this method outperforms all existing trojan detection techniques when compared to previous approaches.
- The paper provides a comprehensive analysis of various data partitioning methods and the impact of the number of partitions.
- The concept of partitioning the data within the trojan class is a novel contribution to the field.

**Weaknesses:**

Weaknesses:

- The paper lacks an in-depth analysis of each design component.
- The writing could be improved for better clarity and understanding.

**Questions:**

Comments:

I am not an expert in trojan injection due to the rapid developments in the field. Nevertheless, I have some questions and concerns about the paper, which I've outlined below:

- Your paper mentions 'unbounded' and 'under-bounded' as key motivations. I understand the 'unbounded' part, but the 'under-bounded' aspect isn't clear to me. Could you explain why "neglecting the combination of triggers" makes adversarial poisoning more easily detectable?

- I noticed the ablation studies in data partitioning, but I didn't find an ablation study regarding the loss function. Are all components equally critical in evading trojan detection? For instance, the 'Dynamic loss' term, which ensures a trojan's effectiveness for a specific partition, what happens if you remove it? Would it expose the trojan?

- Can explicit/implicit "data partitioning" be combined with other trojan injection methods such as WaNet? It seems that the trojan patterns injected into each data partition resemble the human-designed trojan in BadNet (as given in Figure 15). Is my understanding correct?

In addition to these questions, I believe that overall, the paper is OK. However, I recommend improving the writing by highlighting your key ideas and incorporating essential analyses from the appendix into the main paper to make it self-contained.